# TEST TIME AUGMENTATIONS ARE WORTH ONE MILLION IMAGES FOR OUT-OF-DISTRIBUTION DETECTION

## ABSTRACT

Out-of-distribution (OOD) detection is a major threat for deploying machine learning models in safety-critical scenarios. Data augmentations have been proven to be beneficial to OOD detection by providing diverse features. However, previous methods have only focused on the role of data augmentation in the training phase, overlooking its impact on the testing phase. In this paper, we present the first comprehensive study of the impact of test-time augmentation (TTA) on OOD detection. We find aggressive TTAs can cause distribution shifts on OOD scores of In-distribution (InD) data, whereas mild TTAs do not, resulting in the effectiveness of mild TTAs on OOD Detection. Based on the above observations, we propose a detection method that performs a K-nearest-neighbor (KNN) search on mild TTAs instead of InD data. With only 25 TTAs, our method outperforms state-of-the-art methods using the entire training set (1.2 million images) on IMAGENET for OOD detection. Moreover, our approach is compatible with various model architectures and robust to adversarial examples.

## 1 INTRODUCTION

Deep Neural Networks (DNNs) are typically trained in a closed-world assumption. When these models encounter unfamiliar inputs from the open world, they may face out-of-distribution (OOD) samples, which can disrupt the system's normal operation. In safety-critical applications such as autonomous driving Kitt et al. (2010) and healthcare Schlegl et al. (2017), identifying and handling these OOD inputs is crucial. For instance, a self-driving car may fail to detect objects on the road that are not included in the training set, which could lead to an accident.

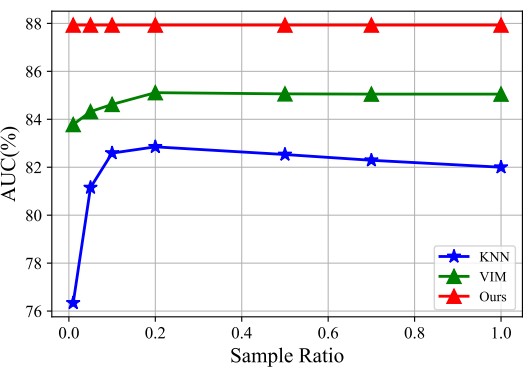

Figure 1: OOD detection performance with different sampling ratios on IMAGENET training set. Our method is InD-independent and thus not affected by the sampling ratio. With only 25 TTAs, our method outperforms KNN Sun et al. (2022) and VIM Wang et al. (2022), which rely on the entire training set (1.2 million images).

To distinguish OOD samples from in-distribution (InD) data, a rich line of OOD detection algorithms have recently been developed. According to the availability of OOD samples, current OOD detection methodologies can be categorized into three categories: OOD Exposure, InD-dependent, and InD-independent Yang et al. (2021c). OOD Exposure involves collecting external OOD samples during training to aid the OOD detector in learning the difference between InD and OOD data. Common methods include OE Hendrycks et al. (2018a), MCD Yu & Aizawa (2019), and UDG Yang et al. (2021b). Although OOD Exposure is simple and effective, it cannot detect unseen OOD data. InD-dependent methods use known InD data as a reference set. For instance, Lee et al. (2018) measure the minimum Mahalanobis distance from the class centroids; KNN Sun et al. (2022) explores the $k-$th nearest neighbor distance between the input sample and the reference set; VIM Wang et al. (2022) uses the reference set for covariance estimation. InD-dependent methods

are influenced by the quantity and quality of InD data, as shown in Figure 1. In contrast, the InD-independent method designs a scoring method based on the output of the model to detect OOD. MSP Hendrycks & Gimpel (2016) and ML Hendrycks et al. (2019a) use the maximum SoftMax and maximum Logit scores to indicate ID-ness, Energy Liu et al. (2020) employs energy-based functions, and ODIN Liang et al. (2017) uses temperature scaling and gradient-based input perturbations. While the InD-independent method is straightforward and user-friendly, its performance requires further improvement.

Currently, numerous studies have demonstrated that data augmentation can enhance the performance of OOD detection. Notable methods include Mixup Zhang et al. (2017), CutMix Yun et al. (2019), and PixMix Hendrycks et al. (2022). However, these methods are mainly applied in the training phase. While He et al. (2022) demonstrated that TTAs can be used for OOD detection, there is a dearth of comprehensive research on the impact of TTAs on OOD detection.

In this paper, we propose an InD-independent OOD detection method based on TTA. First, we present a comprehensive exploration of the effect of TTA on OOD detection. We categorize TTA into In-distribution Augmentation (IDA) and Out-of-distribution Augmentation (OODA) based on their effects on image feature expression. Empirical results reveal that OODA leads to a shift in the distribution of OOD scores, rendering it ineffective for OOD detection. In contrast, IDAs are favorable for OOD detection. Based on these findings, we propose an OOD detection method that boosts KNN with TTA. Specifically, we use the K-th nearest neighbor (KNN) distance between the embedding of the input sample and the generated TTAs to indicate ID-ness, instead of InD data as a reference set. As a non-parametric method, our method does not depend on the information of external OOD data and InD data, and does not modify the model, which is a model-agnostic method. Detection results for common OOD datasets on CIFAR-10 and IMAGENET show that our method outperforms State-Of-The-Art (SOTA) methods. Especially for concurrent KNN Sun et al. (2022), our method only generates 25 TTAs and outperforms the performance of KNN with 1.2 million images as the reference set on IMAGENET (as shown in Figure 1). We summarize our contributions as follows:

1. Our study is the first to investigate the effect of TTA on OOD detection. We classify test-time data augmentations into IDA and OODA and demonstrate that IDA can enhance OOD detection performance. We believe our findings will encourage further research into TTA-based data-efficient OOD detection techniques.

2. We proposed an OOD detection method that employs TTA to improve KNN. Experimental results show that generating as few as 25 TTA samples outperforms SOTA methods achieved by using a reference set of 1.2 million images on IMAGENET.

3. Our method introduces the sequential mask as TTA, and comprehensive evaluations on various OOD detection benchmarks across different model architectures show our method consistently outperforms the SOTA methods. As a plug-and-play method, our method's performance can be further enhanced by incorporating high-quality embeddings. Moreover, our method is also robust to adversarial examples that cause OOD score shifts.

## 2 A CLOSER LOOK AT TEST TIME AUGMENTATION ON OOD DETECTION

Previous research Geiping et al. (2022) firstly classifies data augmentation as aggressive or mild based on whether the augmentation destroys image expression. Empirical results suggest that aggressive data augmentation produces more diverse features, resulting in higher but unstable gains, whereas mild augmentation leads to more stable but weaker gains. Inspired by the above observations, we also classify test-time data augmentation into In-Distribution Augmentation (IDA) and Out-of-Distribution Augmentation (OODA), and investigate its impact on OOD detection:

- **IDA**: TTAs that do not affect the expression of image features, such as horizontal flip (HFlip), gray, small-size center masking, large-size center cropping, and Fourier low-pass filtering.
- **OODA**: TTAs that drastically change the features of the image, such as vertical flip (VFlip), rotation, ColorJitter, Invert, and Fourier high-pass filtering.

First, our investigation revealed that IDA and OODA have distinct effects on the OOD score distribution. Figure 2 illustrates the alterations in the distribution of OOD scores Liu et al. (2020) for both

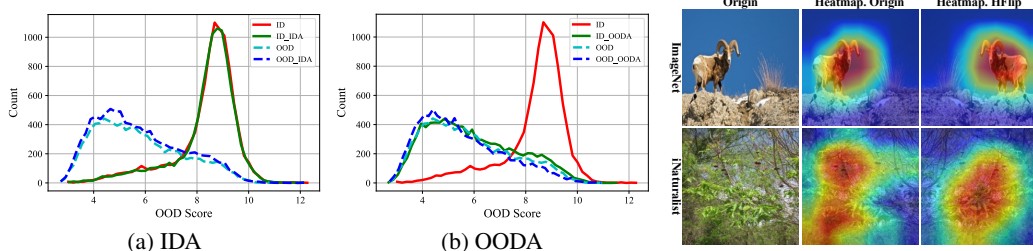

Figure 2: The influence of IDA and OODA on the distribution of OOD Score.

Figure 3: IDA changes the heatmap of OOD, but not the InD.

| TTA | | OOD Datasets | | | | | | |
|---|---|---|---|---|---|---|---|---|
| | | Cifar100 | SVHN | Texture | Places365 | iSUN | LSUN | Average |
| IDA | Hflip | 87.93 | 95.13 | 88.92 | 90.39 | 95.84 | 98.33 | _92.76_ |
| | Gray | 86.77 | 92.49 | 87.38 | 88.71 | 93.42 | 96.75 | 90.92 |
| | CenterMask | 87.43 | 95.07 | 87.89 | 88.49 | 94.24 | 97.99 | 91.85 |
| | CenterCrop | 87.17 | 95.27 | 89.10 | 90.24 | 95.77 | 98.06 | 92.60 |
| | Fourier Low Pass | 87.02 | 94.40 | 89.27 | 90.70 | 96.88 | 98.08 | 92.73 |
| | Hflip + Gray | 87.63 | 95.21 | 88.93 | 90.24 | 95.71 | 98.43 | 92.69 |
| | Hflip + Gray + CenterMask | 88.37 | 95.24 | 88.97 | 90.11 | 95.48 | 98.37 | _92.76_ |
| | Hflip + Gray + CenterMask + CenterCrop | 88.80 | 94.78 | 89.55 | 90.69 | 95.91 | 98.12 | **92.97** |
| OODA | Vflip | 55.88 | 46.14 | 42.71 | 61.53 | 59.89 | 62.06 | 54.70 |
| | Rotate | 53.55 | 50.55 | 45.88 | 61.54 | 58.83 | 58.38 | 54.79 |
| | ColorJitter | 65.87 | 61.88 | 61.03 | 70.80 | 69.05 | 70.65 | 66.55 |
| | Invert | 73.94 | 77.58 | 68.42 | 77.15 | 77.33 | 83.02 | 76.24 |
| | Fourier High Pass | 59.24 | 53.38 | 48.91 | 71.84 | 63.96 | 64.16 | 60.25 |

Table 1: OOD Detection Performance of TTAs on CIFAR-10. The detection performance of IDA is much higher than that of OODA, and using multiple IDAs leads to the optimal performance. See Appendix for results on IMAGENET.

InD and OOD data resulting from IDA and OODA. Our observations reveal that IDA has a negligible effect on the score distribution of InD data, while slightly modifying the distribution of OOD data. In contrast, OODA induces a distribution shift in InD data, making it resemble the distribution of OOD.

Then based on the above findings, we conduct a simple method for OOD detection by comparing output consistency between input samples and their augmentations. The results in Table 1 show that IDA can effectively detect OOD data. Moreover, using multiple IDAs and selecting the one with the highest similarity can further improve the detection performance. In contrast, OODA cannot be used for OOD detection, as it causes the score distribution of InD and OOD data to become similar.

**Why IDA is effective for OOD Detection?** We provide a visual explanation of why IDA is beneficial for OOD detection. We enhance Grad-CAM by modifying the weight computation of feature maps, using a global average of the gradients backpropagated from the Energy score. Figure 3 illustrates the visualization outcomes of InD and OOD data, as well as their IDA results. It can be observed that the OOD heatmap is noticeably affected by IDA, while the InD heatmap remains unaffected, which explains why IDA can be utilized for OOD detection in Table 1.

*Takeaway*: *In contrast to OODA, IDA has the ability to generate distinguishable heatmap differences between InD and OOD data, making it suitable for OOD detection. Furthermore, using multiple IDAs and selecting the most similar can further improve detection performance.*

## 3 METHOD

**Design Objective.** This work considers the classifiers trained on InD data that may encounter OOD samples. Unlike previous methods Sun et al. (2022); Lee et al. (2018), *our goal is to design an effective OOD detection method that requires neither OOD data nor prior knowledge of InD data.* Our approach aims to explore the relationship between a sample and its TTAs, and exploit this for OOD detection. Notably, our method does not alter any component of the classifier, including the architecture and trained weights, making it a model-agnostic plug-and-play detector that can seamlessly integrate with different model architectures.

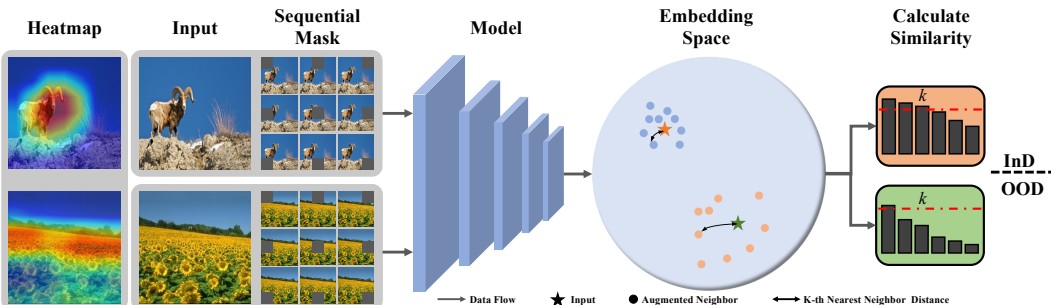

Figure 4: Overview of our method for OOD detection. We first perform a sequential mask for the input image. Next, the input image and corresponding TTAs are fed into the model to obtain embeddings. Then the $k-$th largest similarity between the input image and the TTAs embedding is selected as the ID score. If the score exceeds the threshold, it is detected as InD.

**Core Idea.** Our core idea is to construct a scoring function for OOD detection by utilizing the relationship between samples and their TTAs. Unlike KNN that perform a nearest neighbor search in the feature space of the entire training data, our approach focuses on searching within the local neighborhood of input samples provided by TTAs. Therefore, our method is data-efficient and InD-independent. However, our method requires some IDA that is effective for OOD detection. Therefore, selecting an appropriate TTA strategy becomes crucial for the success of our method.

**TTA Strategy.** According to the conclusion in Sec. 2, the key to improving the performance of OOD detection lies in finding a wide range of effective IDAs. However, the number of common IDAs is limited and incorporating multiple augmentations with varying styles does not lead to improved detection performance, as shown in Table 1, where the performance of Hflip and Gray is weaker than Hflip alone. Inspired by Dosovitskiy et al. (2020), we propose a novel TTA strategy called Sequential Mask, which masks images sequentially, resulting in the generation of a substantial number of similar IDAs. Figure 5 shows the detection performance when utilizing varying numbers of masked images as a reference set. Note that the mask size is 8x8 on CIFAR-10 and 44x44 on IMAGENET. The results demonstrate a clear trend that the detection performance gradually improves with an increasing number of IDAs obtained through sequential mask.

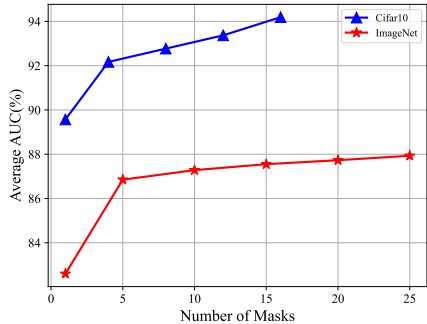

Figure 5: OOD detection performance with different number of masks. The detection performance improves as the number increases.

**Framework.** Figure 4 depicts the overview of our method. We explore the effectiveness of the K-nearest neighbor search in TTAs of input samples for OOD detection. Given an input sample ($x$), multiple TTAs ($x^*$) are generated by sequential mask at first. Then, both the input sample and TTAs are fed into the target model, obtaining embeddings for the input sample ($z$) and its corresponding TTAs ($z^*$). Next, calculate the similarities between $z$ and $z^*$. Finally, the similarities are ranked and the $k-$th largest similarity is selected to indicate ID-ness, which is used to determine whether the input is OOD by a threshold-based criterion as follows:

$$S(z, k) = \mathbf{1}\{-Sim_k(z, z^*) > \lambda\} \qquad (1)$$

where $Sim_k(z, z^*)$ is the cosine similarity to the $k$-th nearest neighbor, and $\mathbf{1}\{\cdot\}$ is the binary indicator function. Typically, the threshold $\lambda$ is selected to ensure accurate classification of the majority of ID data (e.g., 95%). The thresholds are independent of OOD data. The $k$ is selected using the validation method in Hendrycks et al. (2018b). Compared to earlier methods, our method has several compelling advantages:

1. **InD Independent**: Our method does not necessitate any prior knowledge of the InD data. This stands in contrast to KNN Sun et al. (2022) and Mahalanobis distance Lee et al. (2018), and

| Methods | OOD Datasets | | | | | | | | | | | | | |
| | Cifar100 | | SVHN | | Texture | | Places365 | | iSUN | | LSUN | | Average | |
| | F↓ | A↑ | F↓ | A↑ | F↓ | A↑ | F↓ | A↑ | F↓ | A↑ | F↓ | A↑ | F↓ | A↑ |
|---|---|---|---|---|---|---|---|---|---|---|---|---|---|---|
| MSP | 56.29 | 88.11 | 40.67 | 94.36 | 48.74 | 91.13 | 51.96 | 89.24 | 37.80 | 94.03 | 28.59 | 95.91 | 44.01 | 92.13 |
| ML | 49.65 | 88.09 | 28.08 | _95.32_ | 41.33 | 91.06 | _42.52_ | 89.87 | 26.36 | 95.21 | 13.55 | _97.58_ | 33.58 | 92.86 |
| Energy | _48.09_ | 88.18 | 25.63 | **95.49** | 39.77 | 91.15 | **40.59** | 90.02 | 24.34 | 95.38 | _11.85_ | **97.79** | _31.71_ | 93.00 |
| ODIN | 50.86 | 82.10 | _23.14_ | 92.69 | 38.44 | 87.07 | 42.99 | 85.58 | **15.33** | _95.82_ | 10.05 | 97.56 | **30.13** | 90.14 |
| VIM | 54.22 | 87.33 | **15.61** | 94.85 | **25.02** | **94.89** | 50.32 | 89.10 | 30.57 | 95.62 | 47.79 | 94.19 | 37.25 | 92.66 |
| KNN | 51.90 | _90.27_ | 35.32 | 95.31 | 40.30 | _93.86_ | 45.88 | _91.19_ | 28.86 | 95.70 | 28.23 | 96.00 | 38.42 | _93.72_ |
| Ours | **47.36** | **90.79** | 30.39 | 95.17 | _37.07_ | 93.50 | 43.23 | **91.29** | _17.46_ | **96.95** | 15.24 | 97.45 | 31.79 | **94.19** |

Table 2: Comparison with competitive OOD detection methods on CIFAR-10. A is AUROC and F is FPR95, ↑ indicates larger values are better and vice versa. The **bolded** values are the best performance, and the _underlined italicized_ values are the second-best performance, the same below.

VIM Wang et al. (2022), which needs InD data for covariance estimation. Therefore, our method's performance remains unaffected by the InD data (see Figure 1), and it is genuinely distributional assumption-free.

2. **OOD-agnostic**: Our testing process does not depend on any knowledge of the unknown data. Instead, we estimate the threshold using only the InD data.

3. **Data Efficient**: Our method uses a small number of TTAs as the reference set instead of the entire training set.

4. **Model-agnostic**: Our testing procedure solely requires the classifier's output and doesn't modify the classifier. This renders our method applicable to a wide range of model architectures, including convolutional neural networks (CNNs) and the more recent Transformer-based ViT model Dosovitskiy et al. (2020).

## 4 EXPERIMENTS

### 4.1 EXPERIMENTAL SETTING

**ID Datasets.** Following the latest OOD benchmark Yang et al. (2022; 2021a), we chose CIFAR-10 Krizhevsky et al. (2009) and IMAGENET Krizhevsky et al. (2017) as the ID datasets. CIFAR-10 consists of 10 classes of 32x32 color pictures, containing a total of 60,000 images, and each class contains 6000 images. Among them, 50000 images are used as the training set and 10000 images are used as the test set. IMAGENET is a large-scale dataset with 1000 classes, its training set contains 1.2 million images and its validation set contains 50,000 images. We resize all images to 224x224.

**OOD Datasets.** Following the prior work on OOD detection, we choose six and four OOD datasets for CIFAR-10 and IMAGENET respectively. For CIFAR-10, the OOD datasets are Cifar100 Krizhevsky et al. (2009), SVHN Netzer et al. (2011), Texture Kylberg (2011), Places365 Zhou et al. (2017), iSUN Xu et al. (2015) and LSUN Yu et al. (2015), with Cifar100 being the near OOD and the rest being the far OOD. For IMAGENET, the OOD datasets used are iNaturalist Van Horn et al. (2018), Places365, SUN Xiao et al. (2010), Texture, where iNaturalist is near OOD and the rest are far OOD.

**Evaluation metrics.** We mainly use the following two metrics to evaluate OOD detection algorithms: 1) FPR95 measures the false positive rate (FPR) at which the true positive rate (TPR) is equal to 95%, a lower score indicates better performance. 2) AUROC measures the area under the receiver operating characteristic (ROC) curve, showing the relationship between TPR and FPR. The area under the ROC curve can be interpreted as the probability that a positive ID example has a higher detection score than a negative OOD example, with higher scores indicating better performance. In this paper, we use AUROC as the main metric.

**Backbones.** We use ResNet18 He et al. (2016) as the backbone for CIFAR-10. The model is trained for 200 epochs, with a batch size of 128. We use the cosine annealing learning rate Loshchilov & Hutter starting at 0.1. We train the models using stochastic gradient descent with momentum 0.9, and weight decay $5^{-4}$. We use a ResNet50 He et al. (2016) backbone with resolution 224x224 for IMAGENET, and use the pre-trained weights from torchvision maintainers & contributors (2016) with a 76.13% accuracy.

**Baseline Methods.** We compare our methods with six baselines that do not require fine-tuning. They are MSP Hendrycks & Gimpel (2016), MaxLogit Hendrycks et al. (2019a), Energy Liu et al. (2020),

| Method | OOD Datasets | | | | | | | | | |
|--------|--------------|--|--|--|--|--|--|--|--|--|
| | iNaturalist | | Places365 | | SUN | | Texture | | Average | |
| | FPR95↓ | AUROC↑ | FPR95↓ | AUROC↑ | FPR95↓ | AUROC↑ | FPR95↓ | AUROC↑ | FPR95↓ | AUROC↑ |
| MSP | 53.43 | 88.01 | 76.49 | 78.23 | 73.74 | 79.83 | 70.73 | 78.59 | 68.60 | 81.17 |
| ML | 48.32 | *91.31* | *73.28* | *81.03* | 66.35 | 84.39 | 60.78 | 84.26 | 62.18 | 85.25 |
| Energy | 50.54 | 90.96 | 74.01 | 80.80 | 65.02 | *84.52* | 58.69 | 84.57 | 62.07 | 85.21 |
| ODIN | *42.12* | 90.95 | **70.38** | **81.28** | *61.89* | 84.40 | 50.74 | 85.52 | *56.28* | *85.54* |
| VIM | 73.56 | 87.12 | 87.25 | 77.50 | 83.68 | 79.23 | *22.93* | **96.60** | 66.85 | 85.11 |
| KNN | 63.89 | 85.60 | 88.84 | 71.65 | 75.46 | 77.90 | **14.27** | *96.47* | 60.62 | 82.91 |
| Ours | **37.10** | **92.55** | 74.88 | 75.81 | **40.10** | **91.82** | 35.37 | 91.54 | **46.86** | **87.93** |

Table 3: OOD Detection Performance on IMAGENET. The labeling is the same as Table 2.

ODIN Liang et al. (2017), VIM Wang et al. (2022) and KNN Sun et al. (2022). VIM and KNN require 50,000 and 200,000 InD data on CIFAR-10 and IMAGENET, respectively. See Appendix for baseline settings.

## 4.2 EVALUATION ON COMMON CIFAR-10 TASK

**Setup.** Our method is conducted on the logit space for CIFAR-10, and the mask size used is 8x8, and the number of neighbors masked is 16. We use $k = 2$ for detection, which is selected from the pool $k = \{1, ..., 16\}$ using the validation method in Hendrycks et al. (2018b). The choice of space source and $k$-value are discussed further in Sec. 4.6.

**Performance.** Table 2 reports the detection performance of our method and SOTA methods on CIFAR-10. All methods do not use OOD data. VIM and KNN require the entire training set (50,000 images) as a reference set or for covariance estimation. As a SOTA method, KNN achieved an average performance of 93.72% on CIFAR-10. However, our method achieves an average performance of 94.19% without relying on any ID data, which outperforms existing all methods, especially on the near-OOD dataset (CIFAR-100).

## 4.3 EVALUATION ON LARGE-SCALE IMAGENET TASK

**Setup.** The mask size used for IMAGENET is 44x44, and the number of neighbors masked is 25. We use $k = 4$, which follows the same validation procedure as before. For space sources on IMAGENET, we found that the combination of logit and softmax can achieve the most effective results (as shown in Figure 6).

**Performance.** In Table 3, we compare our method with competitive methods on IMAGENET for four OOD datasets. On the near-OOD one (iNaturalist), we achieve the highest AUROC and the lowest FPR95, showing the superiority of our method on hard tasks. On average performance, our method has 87.93% AUROC, which surpasses the second one by 2.39% with the lowest FPR95.

**Comparison with ID-dependent Methods.** Vim Wang et al. (2022) and KNN Sun et al. (2022) need ID data to calculate OOD scores, so their performance is affected by ID data. In contrast, our method computes the OOD score exclusively through TTA. For each detection, KNN searches the $k-$th nearest neighbor within the reference set (usually the entire training set), while our method only needs to perform distance calculations with generated TTAs, reducing the computational cost. Only generating 25 TTAs, our method outperforms KNN with 1.2 million images as a reference set. Moreover, ID-dependent methods are susceptible to unbalanced data Mani & Zhang (2003), while ours does not.

**Limitations.** According to Table 2 and Table 3, we find that our method is weaker than SOTA methods for detecting texture whether on CIFAR-10 or IMAGENET. We think the reason is that texture images are not sensitive to masking. Hence, how to better detect OOD datasets that are not sensitive to TTA is our future work.

## 4.4 BOOST BY ACTIVATION RECTIFICATION

Our method is based on the similarity of the image with its TTAs, and Ming et al. (2023) shows that embedding quality is the key to distance-based OOD detection methods. Activation rectification (ReAct Sun et al. (2021)) can effectively suppress the high activation values on the feature of OOD

| Method | CIFAR-10 | IMAGENET |
|---|---|---|
| Ours | 94.18 | 87.93 |
| ReAct | 92.66 | 90.80 |
| ReAct+Ours | **94.27** | **91.02** |

Table 4: Boosted with ReAct.

| Method | Adversarial Attacks | | | | OOD AUC |
|---|---|---|---|---|---|
| | FGSM | PGD | C&W | Average | |
| SimCLR (Ours) | 77.63 | 81.33 | 71.43 | **76.80** | 91.29 |
| Mask (Ours) | 66.10 | 83.34 | 45.47 | 64.97 | **94.19** |
| MSP | 86.37 | 22.26 | 79.33 | 62.65 | 92.13 |
| ML | 85.62 | 1.84 | 79.25 | 55.57 | 92.86 |
| Energy | 85.48 | 1.84 | 79.16 | 55.49 | 93.00 |
| ODIN | 88.01 | 6.56 | 79.40 | 57.99 | 90.14 |
| KNN | 22.82 | 71.52 | 50.63 | 48.32 | 93.72 |
| VIM | 58.56 | 83.65 | 63.50 | 68.57 | 92.59 |

Table 5: Robustness of OOD Detection Methods on CIFAR-10.

data. We combine our method with ReAct, and the results in Table 4 show that the combination achieves improved performance.

## 4.5 ROBUSTNESS

Azizmalayeri et al. (2022) indicates that existing OOD detection methods have made great progress, but adversarial examples can shift the OOD score distribution. We evaluate the robustness of different OOD detectors on three common adversarial attacks, whose hyperparameters are given in the Appendix. As shown in Table 5, the projected gradient descent (PGD) attack Madry et al. (2017) causes a shift in the OOD scores of methods based on logit and softmax outputs (MSP, ML, Energy, and ODIN), resulting in a crash in detection performance. For distance-based (KNN) and multi-space sources (VIM) detection methods, they detect PGD fairly well, but suffer performance degradation for the simple attack FGSM (fast gradient sign method) Goodfellow et al. (2014). As for our method, when using the sequential mask as the TTA, the detection performance for the C&W attack Carlini & Wagner (2017) is relatively low. This is possibly due to the fact that the tiny perturbation of the C&W attack is not sensitive to masking. The average detection performance is optimal when using SimCLR's combined augmentations Chen et al. (2020). However, SimCLR is more aggressive, leading to a decrease in the detection performance of OOD.

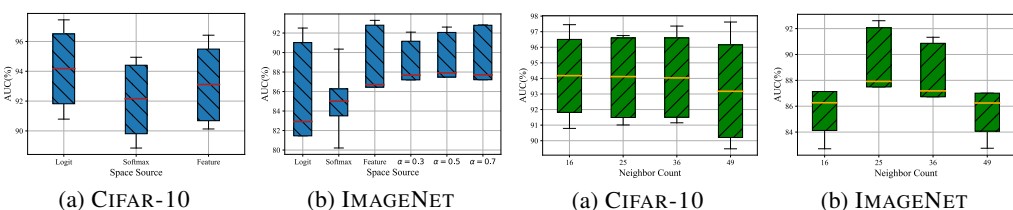

(a) CIFAR-10     (b) IMAGENET     (a) CIFAR-10     (b) IMAGENET

Figure 6: Detection performance using different space source.

Figure 7: Detection performance using different mask size.

## 4.6 ABLATION STUDY

**Source Space.** Wang et al. (2022) points out that the optimal space source for OOD Detection depends on the InD datasets and detection methods. Figure 6 shows the detection performance of our method on CIFAR-10 and IMAGENET using different spatial sources, where the red line represents the average performance for different OOD datasets. It can be seen that on CIFAR-10, logit is the optimal space source, and the combination of logit and softmax each accounting for 0.5 has the best performance on IMAGENET.

**Mask Size.** Figure 7 shows the impact of different mask sizes on detection performance, where the yellow line represents the average performance for different OOD datasets. For CIFAR-10 and IMAGENET, the mask sizes we tested are $\{8, 6, 5, 4\}$ and $\{54, 44, 37, 32\}$ respectively, and the count of generated samples are $\{16, 25, 36, 49\}$. It can be observed that the optimal count of generated samples on CIFAR-10 and IMAGENET is 16 and 25, i.e., the optimal mask size is 8 and 44. Note that the choice of hyperparameters has minimal impact on the detection performance of CIFAR-10. Furthermore, even when using the worst hyperparameter, our method achieves a performance of over 86% on IMAGENET, surpassing the SOTA (85.54%). Therefore, our method is not hyperparameter-sensitive.

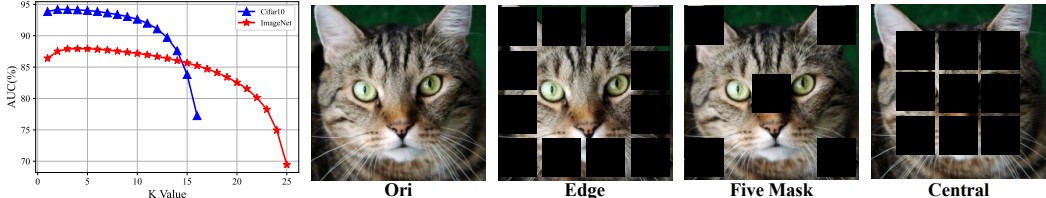

Figure 8: Detection performance of different k.

Figure 9: The Visualization of Different Mask Types.

| Method | OOD Datasets | | | | | | |
|---|---|---|---|---|---|---|---|
| | Cifar100 | SVHN | Texture | Places365 | iSUN | LSUN | Average |
| Sequential Mask | 90.79 | 95.17 | 93.50 | 91.29 | 96.95 | 97.45 | **94.19** |
| Edge Mask | 90.90 | 95.00 | 93.19 | 91.43 | 97.04 | 97.25 | 94.14 |
| Central Mask | 88.00 | 95.41 | 89.72 | 88.78 | 96.04 | 98.15 | 92.68 |
| Five Mask | 90.26 | 95.77 | 92.30 | 91.31 | 96.63 | 98.00 | 94.05 |
| Ten Mask | 90.26 | 95.94 | 90.96 | 91.63 | 96.77 | 98.41 | 94.00 |
| FiveCrop | 86.35 | 93.22 | 90.00 | 90.11 | 95.84 | 97.03 | 92.09 |
| TenCrop | 86.84 | 94.12 | 90.51 | 90.50 | 96.14 | 97.42 | 92.59 |
| SimCLR | 86.00 | 92.37 | 88.78 | 90.01 | 94.16 | 96.43 | 91.29 |

Table 6: Performance of Different TTA Strategies on CIFAR-10.

$k$ **Value in KNN.** In Figure 8, we analyze the effect of $k$. We vary the number of generated samples $k$ from 1 to the maximum on CIFAR-10 and IMAGENET. There are several interesting observations: 1) As $k$ increases, the detection performance exhibits a tendency of slight improvement initially, followed by a sharp decline. 2) When the value of $k$ is small, the gap in detection performance is not large. 3) The optimal $k$ value is 2 on CIFAR-10 and 4 on IMAGENET.

**TTA Strategy.** Our method differs from KNN in that it searches for the nearest neighbors in the samples generated by TTA, rather than in the reference set. Therefore, an appropriate TTA strategy can effectively enhance the detection performance of our method. Table 6 illustrates the detection performance of different TTA strategies on CIFAR-10. The FiveCrop and FiveMask strategies involve cropping and masking the four corners and center of the image, while the TenCrop and TenMask strategies include a horizontally flipped version of the image. It can be observed that sequential masking is the most effective TTA strategy. Furthermore, we conducted a study on the impact of different masking methods on detection performance. Figure 9 presents visualizations of different mask methods. We found that the performance of edge masking is the closest to that of sequential masking. We believe this is because edge masking is milder than center masking, and hence more conducive to OOD detection, which is consistent with the observations in Sec. 2.

| Architectures | OOD Datasets | | | | |
|---|---|---|---|---|---|
| | iNaturalist | Places | SUN | Texture | Average |
| ResNet50He et al. (2016) | 92.61 | 75.78 | 91.88 | 91.39 | 87.92 |
| DenseNet121Huang et al. (2017) | 92.03 | 74.25 | 91.53 | 93.21 | 87.75 |
| WideResNet101Zagoruyko & Komodakis (2016) | 94.41 | 84.28 | 86.36 | 83.80 | 87.21 |
| Vit-b-16Dosovitskiy et al. (2020) | 92.70 | 82.81 | 84.71 | 85.43 | 86.43 |
| Swin-tLiu et al. (2021) | 88.95 | 81.54 | 82.17 | 81.70 | 83.36 |

Table 7: Performance of Our Method with Different Architectures on IMAGENET.

**Architecture.** Table 7 shows the detection performance of our method on different model architectures. From the table, we can see that our method shows good performance for any OOD dataset, regardless of the model employed. As a plug-and-play approach, our method does not require modification of the model structure and parameters. Therefore, there is no additional cost in switching our method between different model structures. Notably, a small decrease in the detection performance of our method occurs when using swin transfomer as the backbone. We further test the detection performance of baselines when using the swin transformer as the backbone in the Appendix. The results show that the performance of baselines all decline and our method remain optimal.

## 5 RELATED WORK

### 5.1 OOD DETECTION METHODS

**OOD Data Exposure Approach.** Some works collect a bunch of external OOD samples to help OOD detectors better learn ID/OOD differences. Outlier Exposure Hendrycks et al. (2018a) utilizes an auxiliary OOD dataset to improve OOD detection. Lee et al. (2017) use GAN to generate OOD samples that are located near ID samples. Several methods including MCD Yu & Aizawa (2019), NGC Wu et al. (2021), and UDG Yang et al. (2021b) can leverage external unlabeled noisy data to enhance OOD detection performance. Although using external OOD data is a simple and effective approach, how to effectively select additional data and how to prevent the model to overfit the given OOD is still an open problem.

**InD-Dependent Approach.** Some InD-dependent methods require InD data as a reference set. Lee et al. (2018) measures the minimum Mahalanobis distance of class centroids, KL-Matching Hendrycks et al. (2019a) computes the minimum KL divergence between softmax and the mean class conditional distribution, and KNN Sun et al. (2022) performs a K-nearest neighbor search on the reference set. VIM Wang et al. (2022) uses InD data to estimate the covariance of features to analyze the main space of features. Another part of the InD-dependent methods requires InD data for training. ConfBranch DeVries & Taylor (2018) builds an additional branch from the penultimate layer to estimate confidence scores. CSI Tack et al. (2020) explores the effectiveness of OOD detectors against learned objectives. MOS Huang & Li (2021) uses priors on supercategories to perform hierarchical OOD detection. VOS Du et al. (2022) produces better energy scores with the support of synthetic virtual outliers. The high performance of InD-dependent methods depends on the quantity and quality of InD data.

**InD-Independent Approach.** InD-independent methods attempt to perform OOD detection by devising scoring functions. MSP Hendrycks & Gimpel (2016) and ML Hendrycks et al. (2019a) directly use the maximum SoftMax score and maximum logits score to detect OOD. ODIN Liang et al. (2017) uses temperature scaling and gradient-based input perturbation. Energy Liu et al. (2020) uses energy-based functions. GRAM Sastry & Oore (2020) computes the gram matrix within hidden layers. DICE Sun & Li (2022) performs weight sparsification in the last layer. GradNorm Huang et al. (2021) focuses on gradient statistics. ReAct Sun et al. (2021) uses rectified activations.

### 5.2 AUGMENTATION FOR OOD DETECITON

Some works have observed that regularizing the model during the training phase using data augmentation will help to better estimate the uncertainty. Mixup Zhang et al. (2017) mixes samples by pair, and AugMix Hendrycks et al. (2019b) mixes samples with their augmentations. CutMix Yun et al. (2019) replaces cut regions in a sample with patches from another image, and PixMix Hendrycks et al. (2022) combines images through additive or multiplicative fusion with additional mixing datasets. YOCO Han et al. (2022) crops images both vertically and horizontally, then mix them in pairs. Mohseni et al. (2021) search for the optimal combination of augmentations through reinforcement learning. Geiping et al. (2022) systematically studied the effect of data enhancement in the training phase on OOD generalization. They found that aggressive augmentations result in more diverse features, while mild augmentations lead to more consistent features. As a result, aggressive augmentations provide a higher but unstable gain, whereas mild augmentations yield a lower but more stable gain. However, the research on the impact of TTA on OOD detection remains elusive.

## 6 CONCLUSION

This paper presents the first systematic study on the impact of TTA for OOD detection and demonstrates that IDA at test time is beneficial and data-efficient for OOD detection. Furthermore, we propose a new TTA-based OOD detection method, which conducts a K-nearest neighbor search on TTAs. Our method only requires a handful of TTAs and spares the need for InD data as a reference set and external OOD data. Extensive experiments show that our method outperforms the SOTA methods on several OOD detection benchmarks. We hope that our work can inspire future research on data-efficient OOD detection using TTAs. We also do not see any immediate ethical concerns or negative societal impacts from this study.

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

# A  APPENDIX

## A  EXPERIMENTS DETAILS

**Software and Hardware.** All methods are implemented in PyTorch 1.13. We run all the experiments on NVIDIA GeForce RTX-3090 GPU.

**Detains of augmentations.** In Table 1, the mask size of CenterMask is 4x4, and the size of CenterCrop cropped image is 30x30 and then resize to 32x32. Both Fourier high-pass filtering and low-pass filtering preserve 90% of the high-pass or low-pass signals. The angle of rotation is 90. In ColorJitter, brightness is 0.4, contrast is 0.4, saturation is 0.4, hue is 0.1.

**Hyperparameters for Baselines.** For VIM, when feature spaces of dimension $N > 1500$, we set the dimension of the main space to $D = 1000$, otherwise set $D = 512$. For KNN, the dimension of the penultimate feature where we perform the nearest neighbor search is 512 and 2048 on CIFAR-10 and IMAGENET respectively, and we choose $k = 50$ following Yang et al. (2022) for detection.

**Hyperparameters for Adversarial Attacks.** We compare the robustness of our method to adversarial attacks with existing OOD detection methods in Table 5. The attack methods we use are FGSM, PGD, and C&W. Among them, the perturbation budget of FGSM is 0.05 ($\epsilon = 0.05$), and that of PGD and C&W is 8/255 ($\epsilon = 8/255$). The number of PGD attack steps is 50, and the step size is 0.002. The maximum number of iterations for C&W is 1000.

## B  HYPER-PARAMETERS IN AUGMENTATIONS

Table 9 investigates the detection performance of rotation and ColorJitter with large disturbance degree. It can be seen that the detection performance of these two augmentations is poor i.e., both are OODA. Moreover, we investigate the detection performance for different rotation angles in the Table 11. Results show that when the angle is small, the detection performance is higher than when the angle is slightly larger. This matches the intuition. When the rotation angle is small, it does not change the image features and can be regarded as IDA; when the rotation angle reaches a certain number of degrees such that it changes the image features, it becomes OODA.

## C  AUGMENTATION USED IN THE TRAINING PHASE

To test the effect of adding augmentation during the training phase on the detection of IDA and OODA, we design three sets of experiments in Table 10 to compare the detection performance of using horizontal flip and vertical flip as TTA on models trained with horizontal flip, vertical flip and no augmentation. It can be observed that the detection performance of horizontal flip is much better than that of vertical flip on the model trained without augmentation. In addition, the performance of

| TTA | | OOD Datasets | | | | |
|---|---|---|---|---|---|---|
| | | iNaturalist | Places365 | SUN | Texture | Average |
| IDA | Hflip | 80.67 | 73.46 | 73.22 | 70.46 | 74.45 |
| | Gray | 85.89 | 69.18 | 75.03 | 66.67 | 74.19 |
| | CenterMask | 81.58 | 68.68 | 76.11 | 73.25 | 74.91 |
| | CenterCrop | 80.11 | 74.24 | 75.49 | 74.22 | 76.02 |
| | Fourier Low Pass | 82.36 | 71.58 | 73.46 | 71.92 | 74.83 |
| | Hflip + Gray | 82.75 | 74.44 | 74.93 | 71.22 | 75.83 |
| | Hflip + Gray + CenterMask | 82.30 | 73.37 | 76.47 | 72.41 | 76.13 |
| | Hflip + Gray + CenterMask + CenterCrop | 83.63 | 73.97 | 76.76 | 73.33 | **76.92** |
| | Hflip + CenterMask | 80.97 | 71.81 | 76.96 | 72.88 | 75.66 |
| | Hflip + CenterCrop | 83.13 | 74.46 | 75.23 | 72.53 | 76.34 |
| | Gray + CenterMask | 80.19 | 70.75 | 77.99 | 73.18 | 75.53 |
| | Gray + CenterCrop | 82.28 | 72.97 | 76.70 | 73.36 | 76.33 |
| | CenterMask + CenterCrop | 81.27 | 71.75 | 77.75 | 73.95 | 76.18 |
| | Gray + CenterMask + CenterCrop | 81.81 | 72.12 | 78.16 | 73.98 | 76.52 |
| OODA | Vflip | 43.42 | 63.61 | 75.30 | 55.72 | 59.51 |
| | Rotate | 43.38 | 63.61 | 75.26 | 55.46 | 59.43 |
| | ColorJitter | 70.28 | 59.21 | 71.80 | 57.91 | 64.80 |
| | Invert | 76.55 | 58.89 | 82.01 | 62.40 | 69.96 |
| | Fourier High Pass | 84.37 | 66.37 | 72.08 | 56.41 | 69.81 |

Table 8: OOD Detection Performance of TTAs on IMAGENET. The detection performance of IDA is much higher than that of OODA, and using multiple augmentations leads to the optimal performance.

| InD Dataset | Rotate | | | ColorJitter | | | |
|---|---|---|---|---|---|---|---|
| | 90 | 180 | 270 | 0.1,0.1 0.1,0.1 | 0.2,0.2 0.2,0.1 | 0.3,0.3 0.3,0.1 | 0.4,0.4 0.4,0.1 |
| Cifar10 | 54.79 | 54.83 | 54.64 | 68.81 | 68.32 | 67.55 | 66.55 |
| ImageNet | 59.43 | 63.28 | 63.49 | 64.79 | 64.80 | 64.81 | 64.80 |

Table 9: Detection performance of OODA under different parameters. For ColorJitter, the 4 numbers represent brightness, contrast, saturation and hue.

vertical flipping is improved on the model trained with vertical flipping. However, it is still weaker than the performance of horizontal flip.

Therefore, since our approach is to compare the output similarity of samples and augmentations, adding some kind of augmentation during the training phase will make this augmentation more like IDA, but it will still not perform as well as a deterministic IDA (e.g., horizontal flipping). Furthermore, since we are using multiple augmentations with K-nearest neighbor search, adding some OODAs will only slightly decrease the overall performance.

## D   OOD DETECTION PERFORMANCE OF TTAS ON IMAGENET

We compare the OOD detection performance of different augmentations when CIFAR-10 is the InD dataset in Sec. 2. Table 8 shows the OOD detection performance of different augmentations on the large-scale dataset (IMAGENET). Consistent with the results in Sec. 2, the detection performance of IDA is much higher than that of OODA, which proves that the division of IDA and OODA is based on whether to destroy common features, and does not depend on the target dataset. Moreover, the detection performance is further improved when a $k$-nearest neighbor search is conducted on multiple augmentations.

## E   VISUALIZATION

Sec. 2 shows that horizontal flipping can cause a difference between the heatmaps of InD and OOD data. To further demonstrate the impact of IDA and OODA on image features, we show the heat maps of common IDAs and OODAs on large-scale datasets in Figure 10. The visualization results of CIFAR-10 are not shown because its resolution is too low. It can be observed that OODA has a great influence on the features of both InD and OOD data. IDA will not change the high thermal area of

| Training Augmentation | Hflip | Vflip |
|---|---|---|
| Hflip | 92.76 | 54.70 |
| No Aug | 90.09 | 56.18 |
| Vflip | 89.58 | 78.23 |

Table 10: Augmentation used in the training phase

| InD Dataset | Rotate Degree | | |
|---|---|---|---|
| | 5 | 15 | 30 |
| Cifar10 | 92.05 | 85.26 | 46.03 |
| ImageNet | 75.85 | 68.40 | 54.09 |

Table 11: Detection Performance of Rotation with different degrees.

InD, while OOD will be affected by IDA. Based on the observation of a large number of visualization results, we have obtained the following empirical conclusions:

- OOD data has a larger proportion of high thermal regions than InD data, that is, the useful features of OOD are more dispersed.

- IDAs do not change the high thermal region of InD, but they will change the high thermal region of OOD. And OODAs have an impact on the features of both InD and OOD. Therefore, IDA can be used for OOD detection, and OODA cannot be used for OOD detection.

- No single IDA was able to cause changes in the high thermal regions of all OOD data. Horizontal flip is an effective TTA for OOD Detection, but the third row of Figure 10 (Places365) shows that horizontal flip does not have as much impact on the heatmap as other TTAs.

Based on the above conclusions, we designed Sequential Mask for OOD detection. First, masking is an IDA that can effectively detect OOD. Then, since the useful features of OOD are more dispersed than InD, the features of OOD are more likely to be changed in the masked samples produced by sequential mask. Finally, The sequential mask can generate multiple Masked samples to make up for the inability of a single IDA to maintain high detection performance for all OODs.

Moreover, we visualise the samples with their masked augmentations in Fig. 11 (a), and it can be seen that there may be some kind of "non-ideal" mask that causes the InD and OOD and their enhancements to be far apart. However, the use of multiple IDAs makes the distance between the InD and its nearest neighbour significantly smaller than that of the OOD.

We also show the distribution of embedding similarity between images from different datasets and their 16 IDAs in Fig. 11 (b). It also shows that multiple IDAs will lead to a significant difference in the distribution of embedding similarity between InD and OOD.

## F    DETECTION PERFORMANCE OF BASELINES ON SWIN TRANSFORMER

| AUC(%) | iNaturalist | Places | SUN | Texture | Average |
|---|---|---|---|---|---|
| MSP | 89.94 | 77.93 | 79.65 | 80.57 | 82.02 |
| ML | 89.07 | 73.06 | 75.58 | 79.08 | 79.20 |
| Energy | 84.99 | 67.47 | 70.88 | 76.44 | 74.95 |
| ODIN | 70.57 | 46.30 | 55.13 | 65.47 | 59.37 |
| VIM | 91.34 | 76.44 | 77.52 | 87.54 | 83.21 |
| KNN | 87.59 | 77.18 | 76.49 | 88.28 | 82.38 |
| Ours | 88.95 | 81.54 | 82.17 | 81.70 | **83.36** |

Table 12: Detection Performance of Different OOD Detection Methods on Swin Transformer.

In Table 7, Our method has significant performance degradation only on the Swin Transformer. To further verify whether our method is architecture-sensitive, we tested the detection performance of common OOD detection methods on Swin Transformer in Table 12. It can be observed that all the detection methods show performance degradation on Swin Transformer. In particular, ODIN shows an average performance degradation of 59.37%. While the average performance of our method is 83.36%, which still outperforms all baselines. Therefore, we conclude that it is not that our method is architecture-sensitive, but that there are some architectures (e.g., Swin Transformer) that are not suitable for OOD detection.

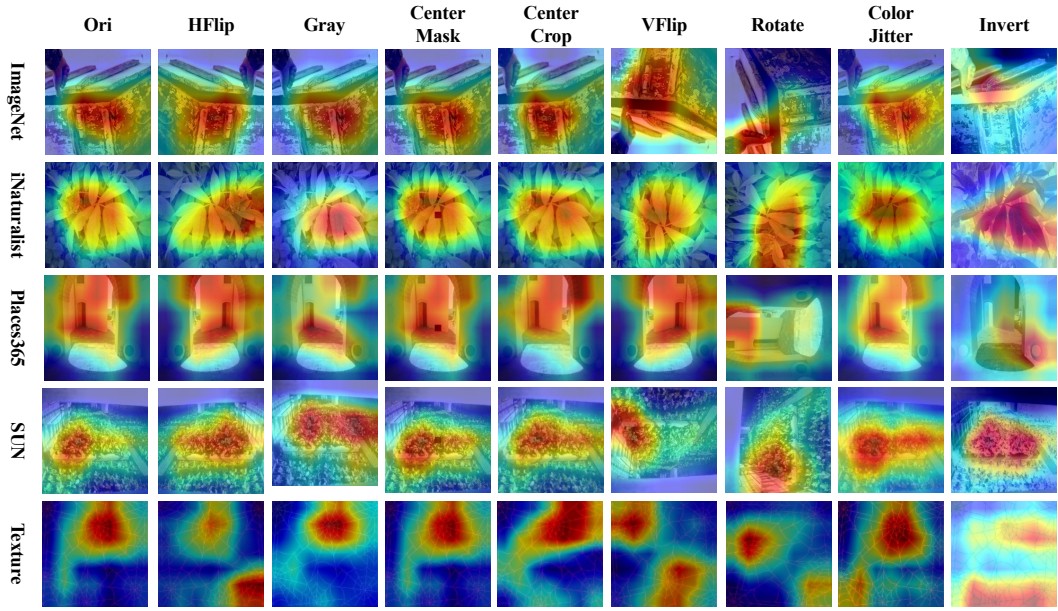

Figure 10: Heatmaps of IDAs and OODAs for InD and OOD. The visualization technology we use is the improved Grad-CAM, which uses a global average of the gradients backpropagated from the Energy score to compute the weights of the feature maps.

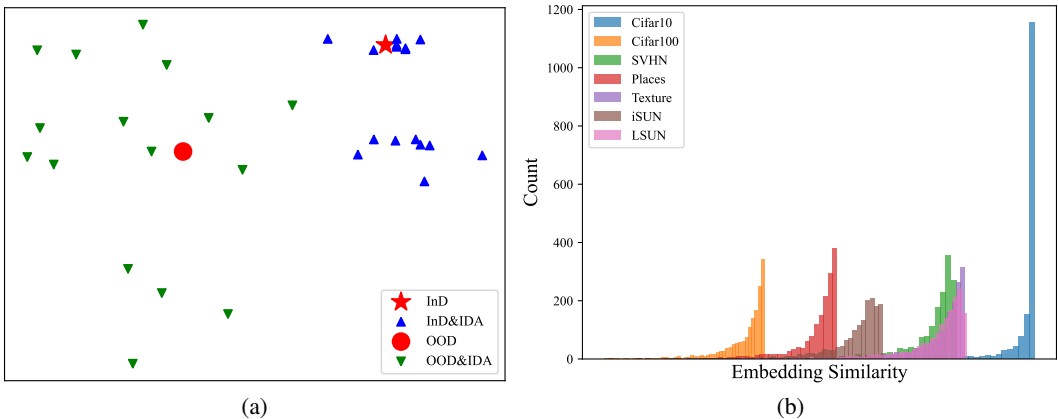

Figure 11: (a) Visualization of embeddings of InD and OOD, as well as their IDAs, It can be observed that the distance between InD and its nearest neighbour is much smaller than OOD. (b) The distribution of embedding similarity between images and their 16 IDAs. It shows that InD (Cifar10) has a higher embedding similarity to its IDAs than OOD.

