# OpenReview forum: "Test Time Augmentations are Worth One Million Images for Out-of-Distribution Detection"
_ICLR.cc/2024/Conference — ICLR 2024 Conference Withdrawn Submission_

### Official Review · Reviewer_xPk1 · 2023-10-28

**Soundness:** 3 good
**Presentation:** 3 good
**Contribution:** 3 good
**Rating:** 5
**Confidence:** 4

**Summary:**

The paper addresses the significant challenge of Out-of-Distribution (OOD) detection in deploying machine learning models for safety-critical applications. It highlights the importance of data augmentations in enhancing OOD detection by diversifying features. While prior research has primarily focused on data augmentation during the training phase, this paper presents the first in-depth exploration of test-time augmentation (TTA) and its impact on OOD detection. The study reveals that aggressive TTAs can cause distribution shifts in OOD scores of In-distribution (InD) data, while mild TTAs do not, showcasing the effectiveness of mild TTAs in OOD detection.

**Strengths:**

The method proposed in the paper is novel and effective.

**Weaknesses:**

1. The lack of comparison with some existing well-performing methods [1][2].
2. There is a lack of consideration of the model architecture's influence; the paper only focuses on one model architecture for each scenario. Table 7 does not compare with existing works. Existing research [3] shows that the choice of model architecture has a significant impact on the performance of detection algorithms. It is essential to compare different algorithms across several models, rather than limiting the comparison to just one model.

I actually enjoyed reading this paper. If the authors can provide reasonable explanations for these issues, I am inclined to increase the score.

[1] Huang R, Geng A, Li Y. On the importance of gradients for detecting distributional shifts in the wild[J]. Advances in Neural Information Processing Systems, 2021, 34: 677-689.
[2] Sun Y, Guo C, Li Y. React: Out-of-distribution detection with rectified activations[J]. Advances in Neural Information Processing Systems, 2021, 34: 144-157.
[3] Zhu Y, Chen Y, Li X, et al. Rethinking Out-of-Distribution Detection From a Human-Centric Perspective[J]. arXiv preprint arXiv:2211.16778, 2022.

**Questions:**

See above.

---

### Official Review · Reviewer_58Yc · 2023-11-01

**Soundness:** 2 fair
**Presentation:** 3 good
**Contribution:** 2 fair
**Rating:** 3
**Confidence:** 3

**Summary:**

This paper presents a simple test time augmentation for detecting OOD.  The idea is to generate an augmented sample employing geometric transformation during test time and compute the distance between the reference image and its augmented images to set the threshold between OODness and IDness.
The experiments are performed on CIFRA-10 and Imangenet and compared with multiple non-parametric OOD detection methods. The results are competitive.

**Strengths:**

OOD detection is an important research problem.

The design of the method is simple and seems effective in the compared baselines.

The paper is generally well-written and easy to understand.

**Weaknesses:**

My concern is how to determine the mild TTAs?  A set of TTAs can be mild for specific data and the same may not be for other data set. For example, rotation for tubular –rotationally invariant – objects may have no effect. However, the same can be a difficult augmentation for various upright objects such as animals, cars, humans etc.

The augmentation is mostly cropping. Is there any guarantee that it generalizes to other domains like text, medical, etc.  The performance on texture recognition shows the limitation of the method.  I appreciate that the authors accept it as a limitation of the work. The solution is clean and effective, but the design of the solution makes the scope of this very narrow.

No theoretical justification is provided.

**Questions:**

Please see the weakness section.

---

### Official Review · Reviewer_jHRr · 2023-11-01

**Soundness:** 2 fair
**Presentation:** 1 poor
**Contribution:** 2 fair
**Rating:** 3
**Confidence:** 3

**Summary:**

This paper focuses on the problem of out-of-distribution (OOD) detection. The authors investigated the impact of test-time augmentation (TTA) on OOD detection and they found that In-Distribution Augmentation (IDA) will not alter the distribution of the In-Domain Data while slightly altering the distribution of the OOD data, which could be used for OOD detection. They further proposed using KNN to search the IDA data to perform better OOD detection.

**Strengths:**

1) Real-world datasets may be polluted by a lot of factors thus out-of-distribution detection is an important problem. So the paper is tackling a meaningful practical problem and any findings would benefit the community.

2) Besides its evaluation of Cifar10 and ImageNet, the paper also includes an evaluation of robustness and ablation studies to check the impacts of different components.

3) The Related Work is well-organized.

**Weaknesses:**

1) The motivation and the problem are not clearly stated. After reading the Introduction, it is very confusing what is the problem and what is the proposed method. Then moving to Section 2, it is even more confusing. Without any definition of what the "OOD Score" is, the authors show figures and tables that use the "OOD Score" as the metric.

2) More deeper explanations are needed for "why IDA is good for OOD detection" as this is the foundation of this proposed method. Currently, the authors only show two images, is it true for other images (images from different classes or other images such as medical imaging or grayscale images)?

3) The "Method" part is also very weak and ambiguous. For example, the "design objective": "This work considers the classifiers..." It sounds like the goal is to build a robust classifier that could be robust to different kinds of corruption. I think more careful description and more detailed explanations are needed for the "Method" part.

4) It is also not clear what the authors mean by "Test Time". Either from the problem setting up, the method, or from the experiments, it is not clear. I think the authors should make this term more clear. Do we need access to the original training dataset? Is it under an unsupervised setting or a supervised setting?

5) What is the dimension of the embedding space? Will it affect the performance? Also, what is the computational overhead of the proposed KNN-based method compared to other methods?

**Questions:**

See [Weaknesses].